# Determination of IgG1 and IgG3 SARS-CoV-2 Spike Protein and Nucleocapsid Binding—Who Is Binding Who and Why?

**DOI:** 10.3390/ijms23116050

**Published:** 2022-05-27

**Authors:** Jason K. Iles, Raminta Zmuidinaite, Christoph Sadee, Anna Gardiner, Jonathan Lacey, Stephen Harding, Gregg Wallis, Roshani Patel, Debra Roblett, Jonathan Heeney, Helen Baxendale, Ray Kruse Iles

**Affiliations:** 1MAPSciences, The iLab, Stannard Way, Bedford MK44 3RZ, UK; jasoniles51@googlemail.com (J.K.I.); raminta.zmuidinaite@mapsciences.com (R.Z.); anna.gardiner@mapsciences.com (A.G.); jonathan.lacey@mapsciences.com (J.L.); 2Laboratory of Viral Zoonotics, Department of Veterinary Medicine, University of Cambridge, Madingley Road, Cambridge CB3 0ES, UK; jlh66@cam.ac.uk; 3The Francis Crick Institute, 1 Midland Road, London NW1 1AT, UK; ch.y.sadee@gmail.com (C.S.); debbie.roblett@crick.ac.uk (D.R.); 4Department of Neuromuscular Diseases, UCL Queen Square Institute of Neurology, Queen Square, London WC1N 3BG, UK; 5The Binding Site Group Ltd., 8 Calthorpe Road, Edgbaston, Birmingham B15 1QT, UK; stephen.harding@bindingsite.com (S.H.); gregg.wallis@bindingsite.com (G.W.); roshani.patel@bindingsite.com (R.P.); 6DIOSynVax, University of Cambridge, Madingley Road, Cambridge CB3 0ES, UK; 7Royal Papworth Hospital NHS Foundation Trust, Cambridge CB3 0ES, UK; hbaxendale@nhs.net; 8NISAD, Sundstorget 2, 252-21 Helsingborg, Sweden

**Keywords:** COVID-19, convalescent plasma, spike protein, nucleocapsid, IgG1, IgG3, predictive profile

## Abstract

The involvement of immunoglobulin (Ig) G3 in the humoral immune response to SARS-CoV-2 infection has been implicated in the pathogenesis of acute respiratory distress syndrome (ARDS) in COVID-19. The exact molecular mechanism is unknown, but it is thought to involve this IgG subtype’s differential ability to fix, complement and stimulate cytokine release. We examined the binding of convalescent patient antibodies to immobilized nucleocapsids and spike proteins by matrix-assisted laser desorption/ionization–time of flight (MALDI-ToF) mass spectrometry. IgG3 was a major immunoglobulin found in all samples. Differential analysis of the spectral signatures found for the nucleocapsid versus the spike protein demonstrated that the predominant humoral immune response to the nucleocapsid was IgG3, whilst for the spike protein it was IgG1. However, the spike protein displayed a strong affinity for IgG3 itself, as it would bind from control plasma samples, as well as from those previously infected with SARS-CoV-2, similar to the way protein G binds IgG1. Furthermore, detailed spectral analysis indicated that a mass shift consistent with hyper-glycosylation or glycation was a characteristic of the IgG3 captured by the spike protein.

## 1. Introduction

The rapidly developing COVID-19 disease syndrome is a delayed event following initial infection of the upper respiratory tract, followed by lower respiratory tissue involvement and a progressive hyper-inflammatory immune response [1]. Somewhat counterintuitively, the marked feature of those requiring hospital treatment is the extremely high titre of antibody and cytokine elevation in those with severe COVID-19 disease [2,3]. However, despite such a massive immune response, the alveoli cells and the blood stream are invaded by viral particles [4].

The humoral response to SARS-CoV-2 is primarily directed towards the nucleocapsid (N protein) and the spike protein (S protein) complex [5]. By the time of ARDS onset, the initial IgM antibody response to the virus has declined and is replaced by IgA and IgG antibodies. IgG is the most abundant class of antibody found in the convalescent plasma of those recovering from COVID-19 ARDS [6] and the onset of ARDS appears to correspond with the time of the antibody class switch to IgG [7] (see Figure 1A). Antibody responses directed at the spike protein and the receptor binding domain (RBD) in particular, have been identified as the main neutralizing component of the SARS-CoV-2 antibody response [8,9,10]. A distinct antibody signature has been linked to different COVID-19 disease outcomes; early spike-specific responses were associated with a positive outcome, while early nucleocapsid-specific responses were associated with a negative outcome and death. Furthermore, the fragment crystallizable (Fc)-associated functions of the antibody response such as antibody-mediated phagocytosis, cytotoxicity and complement deposition are critical for disease resolution [11].

IgG consists of four sub classes, each with structural differences within and adjacent to the hinge region associated with Fc receptor binding and complement activation (see Figure 1C and Table 1). A highly elevated and disproportionate IgG subclass response, dominated by IgG3, has been implicated as a discriminant marker of adverse outcomes in COVID-19 patients [12].

Binding proteins (antibodies in particular) were captured from convalescent patient plasma samples and examined by MALDI-ToF mass spectrometry. The mass spectral signatures of Ig species, IgG subclasses and, in particular, their respective heavy chains, were matched and quantified (Figure 1D). Here we report the comparison of IgG1 and IgG3 (captured by immobilized N-proteins and spike proteins) in relation to disease severity post SARS-CoV-2 infection.

## 2. Results

Post elution from respective antigen-coupled magnetic beads, MALDI-ToF mass spectra were obtained and peaks were recorded. These were matched against reference MALDI-ToF mass spectra obtained from the preparation of purified human serum proteins run under the same reducing and acetic acid pH conditions: human serum albumin (HSA), Transferrin (Merck Life Science UK Ltd., Dorset, UK), IgG1, IgG3, IgA and IgM (Abcam, Discovery Drive, Cambridge, Biomedical Campus, Cambridge, UK).

Data from the pooled polyclonal immunoglobulin isolate showed that IgG1 heavy chains averaged at a peak apex ~51,000 *m*/*z*, IgG3 at ~54,000 *m*/*z*, IgA at ~56,000 *m*/*z* and IgM at ~74,000 *m*/*z*. Human albumin was at 66,400 *m*/*z* (1+) and Transferrin was at ~79,600 *m*/*z*. All were broad heterogenous peaks reflecting glycosylation/glycation and sequence variation. In the protein G, nucleocapsid and spike protein magnetic bead isolates, the dominance of IgG1 was almost always lost, which gave way to IgG3 peak prominence at ~54,000 *m*/*z* and a peak at ~49,000 *m*/*z*. The latter has revealed common peptide sequence fragment matches for IgG following formic acid lysis, but a precise identity confirmation as IgG2 or IgG4 has yet to be established (unpublished data) (see Figure 1D). IgA peaks at ~56,000 *m*/*z* and IgM peaks at ~74,000 *m*/*z* could be detected in some samples, but they were always smaller (at least 1/20th) than the IgG subclasses’ peak intensities. This study focuses on the binding of IgG1 and IgG3 to antigens.

### 2.1. IgG1 Levels and Molecular Mass

Looking at the overall binding from all samples, protein G magnetic beads bound large amounts of IgG1 (peak detected in 96% of samples, median intensity 2870 arbitrary units (AU)), nucleocapsid beads bound very little (peak detected in 33% of samples, median intensity 0AU) and spike protein beads bound a smaller but significant quantity (peak detected in 72% of samples, median intensity 283AU) (see Figure 2). Comparing the three sample groups, protein G bound equally high amounts of IgG1 from sero-negative healthcare workers (HCWs), sero-positive HCWs with mild symptoms and COVID-19 ARDS patient samples (see Figure 3), thus reflecting the polyclonal (non-antigen specific) binding of IgG1 by protein G. Nucleocapsid magnetic beads displayed equally low IgG1 levels in all three sample groups. Hence, there were no significant anti-nucleocapsid IgG1 antibodies, nor binding of polyclonal IgG1 by nucleocapsid proteins. Prefusion complete spike protein displayed the lowest binding of IgG1 in the sero-negative HCW samples (peak detected in 60% of samples, median intensity 186AU), higher levels in the sero-positive HCWs (peak detected in 61% of samples, median intensity 283AU) and significantly higher levels in the COVID-19 ARDS patient samples (peak detected in 92% of samples, median intensity 507AU) (Figure 3 and Figure 4A). Therefore, this finding reflected specific anti-spike IgG1 binding to the immobilized spike protein.

Examining the average molecular mass of the IgG1 bound by or binding to the magnetic beads, although a wide variation of mass (+/−800 *m*/*z*) could be found between individual samples (Figure 2 and Figure 4A), no significant patterns could be detected as characteristics of any of the sample groups.

### 2.2. IgG3 Levels and Molecular Mass

Overall binding showed that protein G could bind IgG3 very rarely (peak detected in 24% of samples, median intensity 0AU) compared to that of the nucleocapsid (peak detected in 51% of samples, median intensity 103AU) and the prefusion spike protein (peak detected in 79% of samples, median intensity 630AU).

Although IgG3 is reported to be bound by protein G, this was low in our samples. Furthermore, Protein G almost completely failed to bind IgG3 in COVID-19 ARDS convalescent patient plasma (peak detected in 0 samples) and only in sero-negative (peak detected in 33% of samples) and sero-positive HCWs (peak detected in 39% of samples). Furthermore, the molecular mass of the IgG3 captured by protein G was uniform and consistent at 54,124–54,137 *m*/*z* and never captured the larger IgG3 seen in the nucleocapsid and spike protein eluants (see Figure 4B and Figure 5).

The nucleocapsid showed an increased capture of IgG3 consistent with disease status: sero-negative HCWs (peak detected in 51% of samples, median intensity 0AU); sero-positive HCWs (peak detected in 52% of samples, median intensity 85AU); and convalescent plasma from COVID-19 ARDS (patients’ peak detected in 59% of samples, median intensity 221AU). This reflects specific IgG3 sample antibodies binding the immobilized SARS-CoV-2 nucleocapsid and not a generic binding of total IgG3 by the nucleocapsid protein (see Figure 4B, Figure 5 and Figure 6). The median mass of the IgG3 heavy chains were 54,267 *m*/*z*.

Spike proteins bound the most IgG3 and this was not consistent with disease status: sero-negative HCWs (peak detected in 83% of samples, median intensity 889AU); sero-positive HCWs (peak detected in 68% of samples, median intensity 443AU); and convalescent plasma from COVID-19 ARDS (patients’ peak detected in 84% of samples, median intensity 623AU). This reflects the fact that immobilized SARS-CoV-2 prefusion spike proteins have a binding affinity for IgG3 antibodies, although anti-spike IgG3-specific binding cannot be excluded and would be masked in this system (see Figure 4B, Figure 5 and Figure 6). A distinctive feature of the IgG3 heavy chains detected in the spike protein capture experiments was the clear preference for a molecular species of around 54,284 *m*/*z* (see Figure 4B, Figure 5 and Figure 6).

## 3. Discussion

The IgG subclasses have unique features associated with complement fixation and Fc receptor binding (see Table 1). The two most potent in this respect are IgG1 and IgG3, but IgG3 normally represents only 2–8% of the immunoglobulin found in serum and plasma, whilst IgG1 accounts for up to 70%. Here, IgG3 was evident as the more dominant IgG subtype in the humoral response to these SARS-CoV-2 antigens, along with another currently unidentified Ig with a heavy chain mass of ~49,000 *m*/*z* that could be IgG4.

The determination as to which Ig isotype is favored during the maturation of an immune response is influenced, in part, by cytokine stimulation of the germinal B cell during Ig heavy chain switching from IgM. Plasma cell formation is induced by interleukin (IL)-21 but modulated by additional cytokines, such as IL-4 promoting switching to IgG and inhibiting switching to IgA. Conversely, IL-10 stimulates IgA production [13].

IgG3 has been reported as the dominant antibody in many viral infections [14]. Thus, it is perhaps unsurprising that IgG3 was the dominant captured subtype of IgG in this study. Interestingly, the nucleocapsid protein had a dominant IgG3 humoral patient response, whilst the spike protein also clearly induced a strong IgG1 humoral patient response.

However, unlike the pattern of the antibody binding to a solid-phase antigen seen in IgG1, the prefusion stabilized spike protein also appeared to have a non-specific binding affinity for IgG3, selectively binding IgG3 from the plasma of sero-negative HCWs, as well as that of patients and sero-positive HCWs. This was similar to our previously reported affinity of the prefusion spike protein for HSA, but of a higher mass form due to advanced glycation end product (AGE) modification (i.e., glycated albumin) [15].

The non-specific IgG3 capture by spike proteins also appeared to be of a higher mass than the low levels of IgG3 captured from the same samples by protein G (Figure 1). Indeed, protein G did not bind this higher molecular mass IgG3 found in the COVID-19 ARDS convalescent patient samples, despite these being the dominant IgG3 molecular mass form captured by the prefusion spike protein.

The mass differences represent only a 0.4% increase of these Ig Hc, but this would fit with the evasion-pathology hypothesis that SARS-CoV-2 binds serum proteins with specific glycan residues or reactive glycation end products [16]. Although immunoglobulins can be similarly AGE as a result of elevated reducing sugars in the blood [16], there is also the strong possibility that the specific variant/inherent glycosylation of IgG3 is the molecular target of this IgG3 binding-coating via prefusion spike protein. Changes in fucosylation and galactosylation of the IgG heavy chain Asparagine N (Asn)-linked glycans have been reported to be a feature of the humoral immune response of those developing ARDS as a result of COVID-19 [17,18,19]. However, there is little, if any, information concerning O-linked glycosylation variation and its effects on antibody bioactivity. IgG3 is unique amongst the IgG subtypes because it has three conserved O-linked glycosylation sites within each of its two defining heavy chain peptides that comprise the extended neck region (see Table 1) [20].

Previous studies in mice and humans suggest that different IgG subclasses show subclass-specific glycosylation patterns [21,22,23,24]. In particular, human IgG3 had less stem fucosylation and branch terminal galactose Asn-linked glycan moieties than IgG1 [24]. A bias towards IgG3 over IgG1 antibodies against the RBD of the spike protein has been reported to be associated with poor prognosis in COVID-19 ARDS patients [12]. Thus, the recent reports of reduced fucose and galactose saccharide residues in anti-SARS-CoV-2 IgG N-linked glycans isolated from patients with severe COVID-19 symptoms could be explained by a change in the ratio of IgG3 to IgG1 antibodies. However, the affinity for a specific higher mass form of IgG3 also points to a different post-translational modification, be it hyper glycosylation, glycation or a mixture of the two processes, which is associated with ARDS arising in individuals infected with SARS-CoV-2. This may prove to be not only a molecular marker of ARDS susceptibility in COVID-19 infected individuals, but also directly related to molecular mechanisms by which SARS-CoV-2 can cause the various vascular and immunological pathologies described [25,26].

## 4. Materials and Methods

### 4.1. Samples

Serum and plasma samples were obtained from Health care workers (HCWs) and COVID-19 patients referred to the Royal Papworth Hospital for critical care during the first wave. NHS healthcare workers working at the Royal Papworth Hospital in Cambridge, UK served as the exposed HCWs cohort (Study approved by Research Ethics Committee Wales, IRAS: 96194 12/WA/0148. Amendment 5). NHS HCWs participants from the Royal Papworth Hospital were recruited through staff email over the course of two months (20 April 2020–10 June 2020) as part of a prospective study to establish sero-prevalence and immune correlates of protective immunity to SARS-CoV-2. Patients were recruited in convalescence either pre-discharge or at the first post-discharge clinical review. All participants provided written, informed consent prior to enrolment in the study. Sera from NHS HCWs and patients were collected between July and September 2020, approximately three months after they were enrolled in the study.

Representative convalescent serum and plasma samples from sero-negative HCWs, sero-positive HCWs and convalescent polymerase chain reaction (PCR)-positive COVID-19 patients were obtained for cross-sectional comparison. The serological screening to classify convalescent HCWs as either positive or negative was done according to the results provided by a CE-validated Luminex assay detecting N-, RBD- and S-specific IgG, a lateral flow diagnostic test (IgG/IgM) and an electrochemiluminescence immunoassay (ECLIA) detecting N- and S-specific IgG. Any sample that produced a positive result by any of these assays was classified as positive. Thus, the panel of convalescent plasma samples (three months post-infection) were grouped in three categories: (A) Sero-negative Staff (*n* = 30 samples). (B) Sero-positive Staff (*n* = 31 samples); and (C) Patients (*n* = 38 samples) [27]. The age, sex and ethnicity of the three groups along with the co-morbidities found in the patient cohort are detailed in Appendix A. As expected, there was an age bias toward older men in the patient cohort (see Appendix A).

### 4.2. Antigen-Coupled Magnetic Beads

Protein G-coupled magnetic beads were purchased from Cytivia Ltd. (Amersham Place, Little Chalfont, Buckinghamshire, UK). Recombinant nucleocapsid and recombinant stabilized complete spike protein magnetic beads were made by Bindingsite Ltd. (Birmingham, UK).

The viral spike protein (S protein) is present on virions as prefusion trimers with the receptor binding domain of the S1 region stochastically open or closed; an intermediary in the S1 region is cleaved and discarded and the S2 undergoes major confirmation changes to expose and then retract its fusion peptide domain [28]. Here, the S protein was modified to disable the S1/S2 cleavage site and maintain the prefusion stochastic confirmation [29].

### 4.3. Semi-Automated Magnetic Bead Capture Processing

The processes of magnetic bead capture, washing, agitation and target binding protein elution can vary dramatically due to damage from overly vigorous mixing and yet, insufficient washing can result in the recovery of large amounts of non-specific binding proteins. To minimize these problems, as well as individual operator variability in the efficiency of target binding protein recoveries, the Crick automated magnetic rack system was employed. This has previously been described in full [15].

### 4.4. Pre-Processing of the Magnetic Beads

1.5 µL microcentrifuge tubes were loaded into the automated magnetic rack. Protein G (GE) and purified nucleocapsid or purified stabilized complete spike magnetic beads (Bindingsite, Birmingham, UK) in their buffer solutions were vortexed to ensure an even distribution of beads within the solution. 10 µL of the appropriate magnetic beads were pipetted into each tube.

100 µL of wash buffer, 0.1% Tween 20 in Dulbecco’s phosphate-buffered saline (DPBS) was pipetted into each tube before resuspending. After mixing for several minutes, the instrument pulled the antigen beads to one side, allowing the wash buffer to be carefully discarded. The wash cycle was repeated three times.

### 4.5. Sample Processing and Binding Fraction Elution from the Magnetic Beads

45 µL of 10× DPBS was pipetted into each of the tubes containing the washed magnetic beads. 5 µL of vortexed neat plasma was pipetted and pump mixed into a tube containing the beads, repeating for each plasma sample. After the resuspension and automated mixing for 20 min, the beads were magnetically collected to one side and the non-bound sample was discarded. Three more wash cycles were conducted using 0.1% DPBS. Subsequently, another 3 wash cycles were conducted using ultra-pure water, discarding the water after the last cycle. 15 µL of recovery solution (20 mM tris(2-carboxyethyl)phosphine (TCEP) (Sigma-Aldrich, Bournemouth, UK) + 5% acetic acid + ultra-pure water) was pipetted into the tubes. The tubes were run alternatively between the ‘Resuspend’ and ‘Mix’ setting for several minutes. After pulling the extracted magnetic beads to one side, the recovery solution was carefully removed using a pipette and placed into a clean, labelled 0.6 µL microcentrifuge tube. This recovery solution was the eluant from the beads and contained the desired proteins.

### 4.6. Sample Analysis by MALDI-ToF Mass Spectrometry

Mass spectra were generated using a 15 mg/mL concentration of sinapinic acid (SA) matrix. The elute from the beads was used to plate with no further processing. 1 µL of the eluted samples were taken and plated on a 96-well stainless-steel target plate using a sandwich technique. The MALDI-ToF mass spectrometer (microflex^®^ LT/SH, Bruker, Coventry, UK) was calibrated using a 2-point calibration of 2 mg/mL bovine serum albumin (33,200 *m*/*z* and 66,400 *m*/*z*) (Pierce^TM^, ThermoFisher Scientific, Waltham, MA, USA). Mass spectral data were generated in a positive linear mode. The laser power was set at 65% and the spectra were generated at a mass range between 10,000 to 200,000 *m*/*z*; pulsed extraction was set to 1400 ns.

A square raster pattern consisting of 15 shots and 500 positions per sample was used to give 7500 total profiles per sample. An average of these profiles was generated for each sample, giving a reliable and accurate representation of the sample across the well. The raw, averaged spectral data was then exported to a text file format to undergo further mathematical analysis.

### 4.7. Spectral Data Processing

Mass spectral data generated by the MALDI-ToF instrument was uploaded to an open source mass spectrometry analysis software mMass™ [30], where it was processed using a single-cycle, Gaussian smoothing method with a window size of 300 *m*/*z*, and a baseline correction with applicable precision and relative offset depending on the baseline of each individual spectra. In software, automated peak picking was applied to produce peak lists, which were then tabulated and used in subsequent statistical analyses.

### 4.8. Statistical Analysis

Peak mass and peak intensities were tabulated in Excel and plotted as graphic comparisons of distributions for each antigen capture and patient sample group. Means and medians were calculated and, given the asymmetric distributions found, non-parametric statistics were applied, such as the Mann-Whitney U test, when comparing differences in group distributions.

## 5. Conclusions

The prefusion spike protein of SARS-CoV-2 has a binding affinity for serum IgG3, along with has, which is mediated via glycation and/or variance in inherent glycosylation. This may be part of an immune evasion/misdirection mechanism. The precise nature of the glycation–glycosylation profile in the susceptibility to, and pathogenesis of, COVID-19-related ARDS requires further study. In addition, humoral immune response reactivity indicates that a nucleocapsid induces a more dominant IgG3 response, whilst a spike protein induces both an IgG1 and IgG3 response. The ratio of IgG1 to IgG3 has been reported by Yates et al. [12] to be important in the development of ARDS and this also requires further investigation.

## Figures and Tables

**Figure 1 ijms-23-06050-f001:**
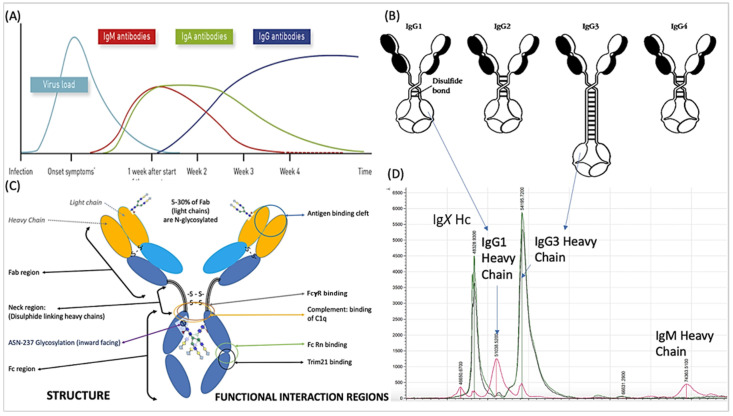
A schematic of the humoral response-immunoglobulin major class switching following SARS-CoV-2 infection and COVID-19 progress is illustrated in (**A**). IgG predominates 3–4 weeks after the onset of symptoms and the major structural and functional domains of IgG are also illustrated. The variation in the neck region of the 4 subclasses of human IgG are shown in (**B**), while (**C**) shows structure and functional interaction regions of immunoglobulin. IgG3 is the largest and its heavy chain (Hc) resolution in MALDI-ToF mass spectra (**D**) is indicated at 54,000 *m*/*z* (IgG3 Hc). Also indicated is the IgG1 heavy chain (IgG1 Hc), which resolves at 51,000 *m*/*z*. The position of the IgM heavy chain peak (IgM Hc) is indicated at 74,000 *m*/*z*. An Ig (IgX Hc), which is thought to be either IgG2 or IgG4 but has yet to be fully identified, was found in patient samples, and is shown at 49,000 *m*/*z*.

**Figure 2 ijms-23-06050-f002:**
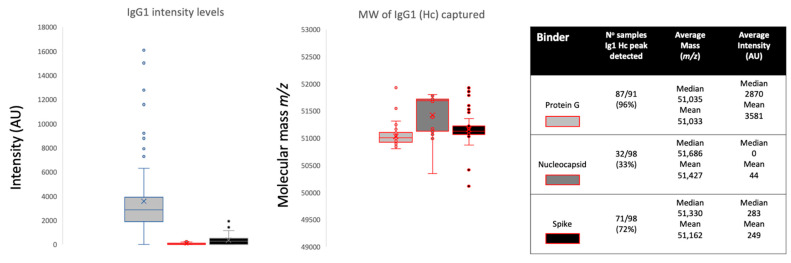
Relative intensities and variance in peak apex molecular mass of IgG1 heavy chains (IgG1 Hc) recovered from the same samples by protein G, nucleocapsid and stabilized spike protein. The non-parametric Kruskal-Wallis statistical test was carried out to test differences between study cohorts. The *p* values (alpha: 0.05) were as follows: intensity *p* < 0.0001 and molecular mass *p* < 0.0001.

**Figure 3 ijms-23-06050-f003:**
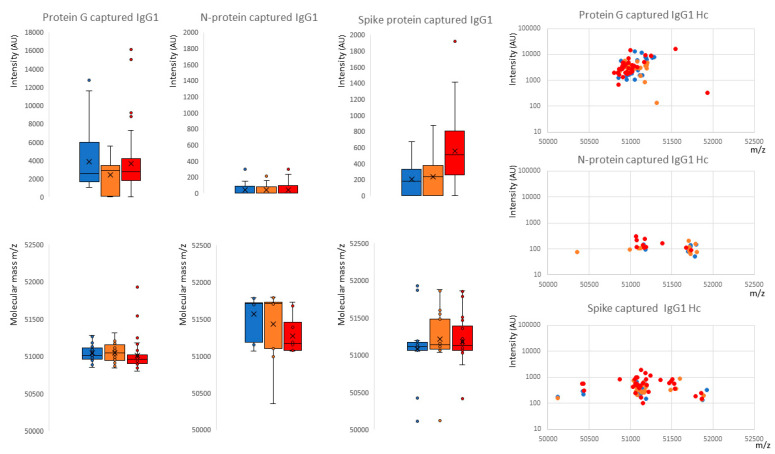
Distribution of intensity for captured and eluted IgG1 heavy chains (IgG1 Hc) and the relative peak molecular mass being bound. The dot plots to the right are intensity versus molecular mass for individual samples; Blue represents data from SARS-CoV-2 sero-negative HCWs, orange from SARS-CoV-2 sero-positive HCWs having recovered from mild symptoms and red sample data from convalescent patients recovering from COVID-19 ARDS. The non-parametric Kruskal-Wallis statistical test was carried out to test differences between study cohorts. The *p* values (alpha: 0.05) were as follows: protein G capture intensity *p* = 0.66, molecular weight *p* = 0.038; protein N capture intensity *p* = 0.84, molecular weight *p* = 0.15; protein S capture intensity *p* = 0.0003, molecular weight *p* = 0.63.

**Figure 4 ijms-23-06050-f004:**
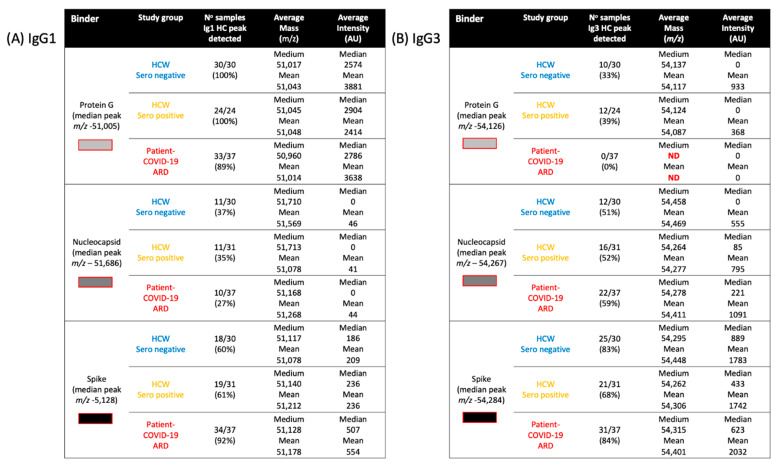
Comparative table of IgG1 (**A**) and IgG3 (**B**) spectral analyses: Detection frequency, intensity and molecular mass for all samples captured by protein G 
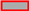
, nucleocapsid 
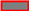
 and prefusion complete spike protein 
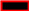
. Further delineation is done by sample infection status: blue represents data from SARS-CoV-2 sero-negative HCWs, orange from SARS-CoV-2 sero-positive HCWs having recovered from mild symptoms and red sample data from convalescent patients recovering from COVID-19 ARDS.

**Figure 5 ijms-23-06050-f005:**
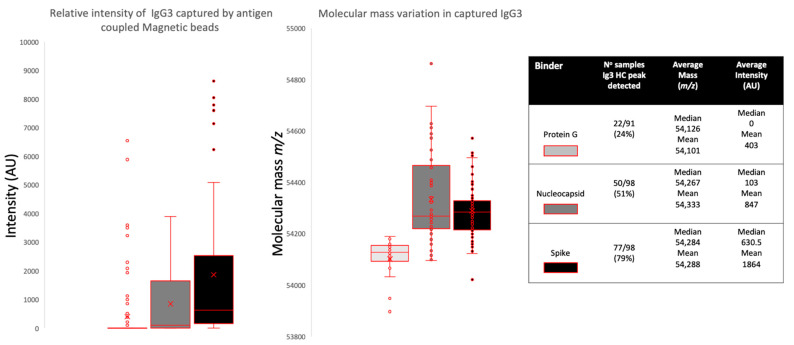
Relative intensities and variance in peak apex molecular mass of IgG3 heavy chains (IgG3 Hc) recovered from the same samples by protein G, nucleocapsid and stabilized spike protein. The non-parametric Kruskal-Wallis statistical test was carried out to test differences between study cohorts, the *p* values (alpha: 0.05) were as follows: intensity *p* < 0.0001, molecular mass *p* < 0.0001.

**Figure 6 ijms-23-06050-f006:**
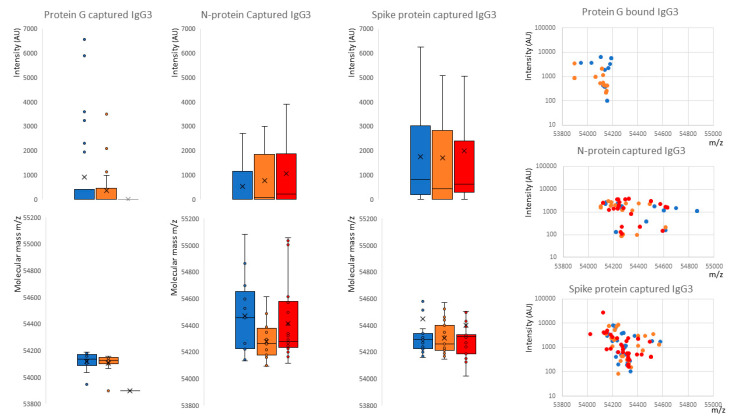
Distribution of intensity for captured and eluted IgG3 heavy chains (IgG3 Hc) and the relative peak molecular masses bound and separated into sample groups. The dot plots to the right are intensity versus molecular mass for individual samples. Blue represents data from SARS-CoV-2 sero-negative HCWs, orange from SARS-CoV-2 sero-positive HCWs having recovered from mild symptoms and red sample data from convalescent patients recovering from COVID-19 ARDS. The non-parametric Kruskal-Wallis statistical test was carried out to test differences between study cohorts; the *p* values (alpha: 0.05) were as follows: protein G capture intensity *p* = 0.61, molecular weight *p* = 0.006; protein N capture intensity *p* = 0.067, molecular weight *p* = 0.19; protein S capture intensity *p* = 0.75, molecular weight *p* = 0.97.

**Table 1 ijms-23-06050-t001:** Descriptive comparison of physical structure and functional biological differences between the four human IgG subclasses.

	IgG1	IgG2	IgG3	IgG4
Relative Serum abundance	43–75%	16–48%	1.7–7.5%	0.8–11.7%
Mean adult serum level (g/L)	6.98 g/L	3.8 g/L	0.51 g/L	0.56 g/L
Half-life (days)	21	21	7 to 21	21
Responses to: Protein	++	+/−	++	++
Polysaccharides	+	+++	+/−	+/−
Allergens	+	(−)	(−)	++
**Total mass ***	146 kDa	146 kDa	170 kDa	146 kDa
Average observed Kappa	23 kDa			
Average observed Lambda	24 kDa			
Average observed Heavy chain	51 kDa	51 kDa	56 kDa	49 kDa
Heavy chain N-linked Glyc sites	1	1	2	1
Heavy chain O-linked Glyc sites	0	0	3	0
Amino acids in hinge region	15	12	62	12
Inter-chain disulphide	2	4	11	2
C1q binding	++	+	+++	−
Receptors binding:				
Fcy-RI	+++	−	+++	+
Fcy-RIIa	++	+/−	++	+/−
Fcy-RIIb/c	++	+/−	++	++
Fcy-RIIIa	+	+/−	+++	+/−
Fcy-RIIIb	+/−	+/−	+	−
FcRn	+++	+++	++/+++	+++
Placental transfer	+++	++	++/+++	+
Protein G binding	++	++	++	+
Protein A binding	++	++	−	++

* Mass as determined by sodium dodecyl sulphate-polyacrylamide gel electrophoresis (SDS-PAGE).

## Data Availability

Compiled summary data can be made available upon request to the corresponding author. Raw mass spectral data from the individual samples will require compiling from archives at MAPSciences and thus, a detailed project proposal justifying the additional resource expenditure necessary to provide this complete data set is required.

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
