# Peer review of "Determination of IgG1 and IgG3 SARS-CoV-2 Spike Protein and Nucleocapsid Binding—Who Is Binding Who and Why?"

_ijms, 2022, doi:10.3390/ijms23116050_

Round 1

Reviewer 1 Report

In “Determination of IgG1 and IgG3 SARS-CoV-2 spike protein and nucleocapsid binding – Who is binding who and why?”  Jason Iles et al. used the nucleocapsid (N) and spike (S) protein magnetic beads to pull down the binding immunoglobins in the plasma samples of convalescent patients. Then those eluted immunoglobins were assessed by matrix-assisted laser desorption/ionization-time of flight (MALDI-ToF) mass spectrometry (MS) to identify the characters of N- and S-binding immunoglobins in the plasma of SARS-CoV-2 negative, convalescence either mild or COVID-19 ARDS. They found that anti-N antibodies are IgG3, but anti-S antibodies seem different from the anti-N. And MS data also showed that the mass shift consistent with hyper-glycosylation or glycation on the anti-S IgG3 antibodies.

First, the severity-associated anti-N, RBD, and S1 IgG subclass skewed toward IgG3 after SARS-CoV-2 infection had been reported ( https://doi.org/10.1016/j.xcrm.2021.100329). So this is not new, and it is ok to use the other methods to confirm these results. However, this manuscript failed to present these major ideas.

Second, based on a mass shift in the MS data, the authors conclude the hyper-glycosylation or glycosylation of the SARS-CoV-2 specific IgG3 antibodies. However, compared with non-SARS-CoV-2 specific IgG3 antibodies, the major differences are the antigen-binding region on the variable region of the light chain and the heavy chain of immunoglobulins. In contrast, most glycosylation sites in the heavy chain are still remarkably preserved. In addition, the authors did not do more experiments to provide more data to support their hypotheses, such as de-glycosylation of binding-IgG3.

Other major issues:

  1. Study cohorts and plasma samples
    1. Missing a lot of information on study cohorts, e.g., the sample collection time (after diagnosis or symptom-on-set, since the antibody response is time-dependent), classified the cohorts, and the concept of COVID-19 ARDS.
    2. Missing subjects’ demography information, even in a pre-printed paper.
  1. The experiment methods and results are very confusing
    • Protein G should bind all isotypes of total IgG in the serum samples, as shown in Table1. If you run the same plasma samples pull-down by protein G beads, the IgG1 and IgG3 should be more than the amount of anti-N and anti-S IgG1 and IgG3, but in figure 4, the IgG3 intensity was very low in the protein G group. Which does not make sense.
    • All figures and tables missed the statistically significant test results. Without the p values of testing, how can we get any differences between each study cohort?
    • Figure 2 and figure 4 should come from the same MS experiment and show the same measurements, one for IgG1, and another for IgG3. It did not make sense to separate them.
    • Figure 3 and Figure 5 missed the group labeling on the figure section, panel labelings, and x and y-axis labeling on the right-side plots.
    • Table 2 is not a scientific table and needs to be revised.
    • The methods description needs to be revised in a precise and scientific way.
  2. Again, in conclusion, this study did not show any evidence for the “spike protein binding to IgG3 is mediated via glycation and/or variance in inherent glycosylation”, but not other IgG3 antibodies.

In general, this study did not have a good experiment design with valid controls. The results they showed could not support their hypothesis. It could be considered with significant revisions and do more experiments.

Author Response

First, the severity-associated anti-N, RBD, and S1 IgG subclass skewed toward IgG3 after SARS-CoV-2 infection had been reported ( https://doi.org/10.1016/j.xcrm.2021.100329). So this is not new, and it is ok to use the other methods to confirm these results. However, this manuscript failed to present these major ideas.

A:

This manuscript is presenting the confirmatory data based on a different technique, which is more robust and economically viable to run, for both R&D and clinical applications. We support conclusions presented in Yates et al., 2021.

We have also changed the preprint reference to the published manuscript.

Second, based on a mass shift in the MS data, the authors conclude the hyper-glycosylation or glycosylation of the SARS-CoV-2 specific IgG3 antibodies. However, compared with non-SARS-CoV-2 specific IgG3 antibodies, the major differences are the antigen-binding region on the variable region of the light chain and the heavy chain of immunoglobulins. In contrast, most glycosylation sites in the heavy chain are still remarkably preserved. In addition, the authors did not do more experiments to provide more data to support their hypotheses, such as de-glycosylation of binding-IgG3.

A:

We have concluded the hyper-glycosylation or glycation (not glycosylation) as this is a different phenomenon, independent from the controlled enzymatic modification. Furthermore, IgG3 subclass have a unique neck region with O-linked glycosylation sites as described in the lines 228-231. Thus glycosylation of IgG subclasses  are not remarkably conserved only one site is common (see figure and table).

We agree, that further experiments of de-glycosylation could be performed, however it was outside the scope of this article and the study.

Other major issues:

Study cohorts and plasma samples

Missing a lot of information on study cohorts, e.g., the sample collection time (after diagnosis or symptom-on-set, since the antibody response is time-dependent), classified the cohorts, and the concept of COVID-19 ARDS.

A: The analysed samples were convalescent plasma and sample collection was specified as follows “Patients were recruited in convalescence either pre-discharge or at the first post-discharge clinical review.”

The ARDS was thoroughly described in the first of the series published article https://doi.org/10.3390/ijms23084126

Missing subjects’ demography information, even in a pre-printed paper.

A:

Demographic information has been added as a supplementary material.

The experiment methods and results are very confusing

Protein G should bind all isotypes of total IgG in the serum samples, as shown in Table1. If you run the same plasma samples pull-down by protein G beads, the IgG1 and IgG3 should be more than the amount of anti-N and anti-S IgG1 and IgG3, but in figure 4, the IgG3 intensity was very low in the protein G group. Which does not make sense.

A:

The section 2.2 reports the findings of the IgG3 binding to the three antigens and the observed difference among the cohorts. It is reported that the molecular mass of the IgG3 captured by Protein-G was uniform and consistent at 54,124 – 54,137m/z and never captured the larger IgG3 seen in nucleocapsid and spike protein eluants (lines 163 - 165).

These are indeed interesting findings and thus we are reporting them.

All figures and tables missed the statistically significant test results. Without the p values of testing, how can we get any differences between each study cohort?

A:  Values for the non-parametric Kruskal-Wallis test were added for Figures 2, 3, 4 & 5, providing test results for all the cohorts tested.

Figure 2 and figure 4 should come from the same MS experiment and show the same measurements, one for IgG1, and another for IgG3. It did not make sense to separate them.

A:

The results come from the same MS experiment, the analysis of the same spectra at the different MS region, specific for IgG1 and IgG3 respectfully.

The main reason for separating the figures was to present IgG1 and IgG3 as this is how the authors chose to separate the analysis focusing on these specific subtypes of IgG as per subsections in the manuscript.

Figure 3 and Figure 5 missed the group labeling on the figure section, panel labelings, and x and y-axis labeling on the right-side plots.

A:

Figure 3: Group labelling are described in the caption of the figures. Axis labelling have been corrected.

Figure 5: Group labelling are described in the caption of the figures. Axis labelling have been corrected.

Table 2 is not a scientific table and needs to be revised.

A:

We disagree with a comment and feel that the table is appropriate, depicting the intensity, binding frequency and molecular mass of the three cohorts in each of the three antigen groups.

The methods description needs to be revised in a precise and scientific way.

A:

It is not clear where reviewer disagrees with the methods or what specific section of the methods such as spectral analysis or the data acquisition by the MALDI-ToF MS is inane.

The used materials have the necessary source and description, methods previously published were thoroughly described in those publications and referenced accordingly.

Again, in conclusion, this study did not show any evidence for the “spike protein binding to IgG3 is mediated via glycation and/or variance in inherent glycosylation”, but not other IgG3 antibodies.

 A:

If the reviewer meant other IgG3 antigens, this is shown in the Figure 4, where molecular variation of the captured IgG3 by three antigens (S, N and G) are shown.

The hypothesis of glycation and/or glycosylation of IgG3 was reviewed and presented in the literature, however due to lack of the sample volume, further tests were not performed.

In general, this study did not have a good experiment design with valid controls. The results they showed could not support their hypothesis. It could be considered with significant revisions and do more experiments.

A:

We again thank reviewer for their comments and hope we were able to answer the raised questions and edited what was necessary in the manuscript.

Reviewer 2 Report

Very interesting work.

Major comments:

1. for fig 1,

(1) the current title not match the legend

(2) for fig 1B, too much irrelevant information making the fig complicated, suggest to simplify the fig, and emphasize the major point

2. for table 1, the author needs to think about what's the major point that would like to make here, and how table 1 could support the major point.

the current organization looks like simply showing some generic data.

3. line 90, where is fig 1 panel D?

4. for fig 3, please label the x and y axises correctly

Author Response

Thank you for your comments, all the figures have been corrected and adjusted accordingly.

1.(1) Panel identification was corrected.

1.(2) This figure demonstrates the variation in the immunoglobulins from different perspectives, including response time to disease, neck region of an IgG subtype, the data acquired from MALDI-ToF and also showing an overall structure of immunoglobulin.

  1. The main point of this table is to show and compare the differences of all immunoglobin subtypes all gathered in one cohesive table.
  2. The panel identification was adjusted.
  3. The axes labelling was corrected and figure reuploaded.

Round 2

Reviewer 1 Report

The authors answered all questions and revised the manuscript based on suggestions.

I agree to accept it in the current format.

Author Response

Thank you.